# Non-Expressed Donor KIR3DL1 Alleles May Represent a Risk Factor for Relapse after T-Replete Haploidentical Hematopoietic Stem Cell Transplantation

**DOI:** 10.3390/cancers15102754

**Published:** 2023-05-13

**Authors:** Nolwenn Legrand, Perla Salameh, Maxime Jullien, Patrice Chevallier, Enora Ferron, Gaelle David, Marie-Claire Devilder, Catherine Willem, Ketevan Gendzekhadze, Peter Parham, Christelle Retière, Katia Gagne

**Affiliations:** 1Etablissement Français du Sang (EFS), F-44011 Nantes, France; nolwenn.legrand@efs.sante.fr (N.L.); perla.salameh@efs.sante.fr (P.S.); maxime.jullien@efs.sante.fr (M.J.); enora.ferron@efs.sante.fr (E.F.); gaelle.david@efs.sante.fr (G.D.); catherine.willem@efs.sante.fr (C.W.); christelle.retiere@efs.sante.fr (C.R.); 2Institut National de la Santé et de la Recherche Médicale (INSERM) UMR1307, CNRS UMR 6075, Centre de Recherche en Cancérologie et Immunologie Integrée Nantes Angers (CRCI2NA), Team 12, F-44000 Nantes, France; patrice.chevallier@chu-nantes.fr (P.C.); marie-claire.devilder@univ-nantes.fr (M.-C.D.); 3LabEx IGO “Immunotherapy, Graft, Oncology”, F-44000 Nantes, France; 4Department of Hematology Clinic, Nantes University Hospital, F-44000 Nantes, France; 5Department of Hematology and HCT, HLA Laboratory, City of Hope, Medical Center, Duarte, CA 91010, USA; kgendzek@coh.org; 6Department of Structural Biology and Microbiology & Immunology, Stanford University School of Medicine, Stanford, CA 94305, USA; peropa@stanford.edu; 7LabEx Transplantex, Université de Strasbourg, F-67000 Strasbourg, France

**Keywords:** KIR3DL1, null alleles, allelic polymorphism, haploidentical HSCT, relapse, leukemia, post-transplant cyclophosphamide

## Abstract

**Simple Summary:**

Natural killer (NK) cells are key cytotoxic effectors against leukemic cells. The polymorphism of killer cell immunoglobulin-like receptor (KIR) genes plays a crucial role in the NK cell repertoire. In particular, different levels of KIR3DL1 expression on the NK cell surface are described, discriminating non-expressed vs. expressed allotypes depending on the KIR3DL1 alleles. KIR3DL1 allelic polymorphism after T-replete haploidentical hematopoietic stem cell transplantation (hHSCT) has not yet been investigated. In this study, we first assessed the extent of non-expressed versus expressed KIR3DL1 allotypes in a cohort of healthy blood donors and then evaluated their clinical impact on relapse incidence after hHSCT. Overall, we would expect that taking KIR3DL1 allelic polymorphism into consideration could help to refine the scores used for HSC donor selection.

**Abstract:**

KIR3DL1 alleles are expressed at different levels on the natural killer (NK) cell surface. In particular, the non-expressed KIR3DL1*004 allele appears to be common in Caucasian populations. However, the overall distribution of non-expressed KIR3DL1 alleles and their clinical relevance after T-replete haploidentical hematopoietic stem cell transplantation (hHSCT) with post-transplant cyclophosphamide remain poorly documented in European populations. In a cohort of French blood donors (N = 278), we compared the distribution of expressed and non-expressed KIR3DL1 alleles using next-generation sequencing (NGS) technology combined with multi-color flow cytometry. We confirmed the predominance of the non-expressed KIR3DL1*004 allele. Using allele-specific constructs, the phenotype and function of the uncommon KIR3DL1*019 allotype were characterized using the Jurkat T cell line and NKL transfectants. Although poorly expressed on the NK cell surface, KIR3DL1*019 is retained within NK cells, where it induces missing self-recognition of the Bw4 epitope. Transposing our in vitro observations to a cohort of hHSCT patients (N = 186) led us to observe that non-expressed KIR3DL1 HSC grafts increased the incidence of relapse in patients with myeloid diseases. Non-expressed KIR3DL1 alleles could, therefore, influence the outcome of hHSCT.

## 1. Introduction

Killer cell immunoglobulin-like receptors (KIRs) play a crucial role in the structure of the natural killer (NK) cell receptor repertoire and the education of NK cells through interactions with self-HLA class I molecules [1,2]. KIRs comprise a family of inhibitory (2DL, 3DL) and activating (2DS, 3DS) receptors clonally expressed on NK cells and some T cell subsets [3]. All HLA-Cw variants function as specific ligands for KIR2DL1/2/3, whereas only HLA-A and HLA-B molecules represent the Bw4 epitope function as ligands for KIR3DL1 [4]. KIR genes exhibit a specific organization, including variations in their content, haplotypes, centromeric (cen) and telomeric (tel) motifs, and allelic polymorphisms [5]. Among the KIRs, the inhibitory KIR3DL1 and its activating KIR3DS1 counterpart are segregated as alleles of the same gene, and there is the possibility that individuals can have multiple copies of KIR3DL1/S1 [6]; thus, they are among the most intriguing KIRs [7]. While the KIR3DL1 gene is present on the A KIR haplotype at a frequency close to 95%, KIR3DS1 is, at present, only on the B KIR haplotype (40%). Thus, the inheritance of KIR3DL1 or KIR3DS1 defines different tel motifs [8]. In addition, an extensive allelic polymorphism with 189 KIR3DL1 and 91 KIR3DS1 alleles has been described and reported in the last IPD KIR database.

KIR3DL1/S1 polymorphism has been correlated with a variety diseases [9,10,11,12,13], with hematopoietic stem cell transplantation (HSCT) outcomes [14,15,16], and with viral infections [17,18,19,20,21]. KIR3DL1 allele polymorphism impacts both KIR3DL1^+^ NK cell phenotype and function [22,23,24,25,26]. Different levels of KIR3DL1 expression on NK cell surface are described, including null (N), low (L), and high (H) allotypes, depending on the KIR3DL1 alleles [23]. KIR3DL1*004 is poorly expressed on the NK cell surface [27,28] and has been associated with improved outcomes for patients with neuroblastoma [29]. KIR3DL1*004 is also reported to be protective against relapse in patients with acute myeloid leukemia (AML) after HLA-matched HSCT [15] and against HIV disease progression [30], which emphasizes the underlying and intriguing function of this frequent null KIR3DL1 allele in Caucasians. In contrast, KIR3DL1*004 is absent from the Japanese population [31] and the Chinese Han population [32].

We previously reported that the KIR3DL1/S1 gene has an impact on HSCT outcomes [33,34] and that the nature of both KIR3DL1 alleles and the KIR3DL1/S1 allele combination is involved in modulating the repertoire of KIR3DL1^+^ NK cells [35]. Notably, we reported a high proportion of individuals who had a null KIR3DL1 allele, such as KIR3DL1*004, L1*019, and L1*054 [35], as well as the induction of KIR3DS1 expression on NK cells by various stimuli in KIR3DS1^+^/KIR3DL1^null^ individuals [36]. The roles of 3DL1*019, first identified in Caucasian individuals [37], and 3DL1*054 [22], which are less frequent than KIR3DL1*004, remain unknown. In this study of a cohort of volunteer blood donors, the distribution of non-expressed vs. expressed KIR3DL1 alleles was assessed using a high-resolution next-generation sequencing (NGS) technology that we developed [38], which was used to characterize the phenotype and Bw4 recognition of the KIR3DL1*019 allotype using specific KIR3DL1 constructs and dedicated KIR3DL1 mutagenesis. We further document the clinical relevance of non-expressed vs. expressed KIR3DL1 alleles to relapse incidence in a local cohort of hHSCT patients.

## 2. Materials and Methods

### 2.1. Cells (PBMCs and Cell Lines)

Peripheral blood mononuclear cells (PBMCs) from 278 healthy adult volunteers were isolated by density centrifugation on Ficoll–Hypaque (Biosera, France). All blood donors were recruited at the Blood Transfusion Center (Etablissement Français du Sang, Nantes, France) and gave informed consent. The preparation and conservation of these biocollections have been declared to the French Research Minister (DC-2014-2340), and the study was approved by the IRB (2015-DC-1). The Jurkat T cell line was cultured in an RPMI 1640 medium containing glutamax and penicillin-streptomycin, supplemented with 10% FBS (all from Life Technologies, Paisley, UK). The KIR3DL1*019^L86S+S182P^ transfected NKL cell line was obtained by stable transfection of NKL by electroporation (Biorad) using pcDEF3-3DL1*019^L86S+S182P^, as described in the “KIR3DL1 constructs” section. HLA-B*15:13 (Bw4^+^), HLA-B*38:01 (Bw4^+^), HLA-B*35:01 (Bw4^−^), and HLA-B*39:01 (Bw4^−^) transfected 721-221 lymphoblastoid EBV-B cells (referred to as Bw4^+^ and Bw4^−^ 221 cells) were used to evaluate degranulation in the KIR3DL1*019^L86S+S182P^ transfected NKL cell line. The KIR3DL1*019^L86S+S182P^ transfected NKL cell line and HLA-B transfected 221 cell lines were cultured in RPMI 1640 medium containing glutamax and penicillin-streptomycin, supplemented with 10% FBS and G418 geneticin (GIBCO, Thermo Fisher Scientific, Waltham, MA, USA) at 2 mg/mL. Mycoplasma tests performed by PCR were negative for all cell lines.

### 2.2. Cohort of T-Replete Haploidentical HSCT Patients

This study analyzed a cohort of 186 adult patients with hematological malignancies who underwent T cell-replete haploidentical hematopoietic stem cell transplantation (hHSCT) with post-transplant cyclophosphamide (PTCy) in the Hematology Department of Nantes University Hospital. Various conditioning regimens were used, including a reduced-intensity TBF regimen [39], a Baltimore-based regimen [40,41,42], and a myeloablative or sequential regimen [40,43]. The source of grafts in all cases was peripheral blood stem cells from a haploidentical donor. Graft versus host prophylaxis consisted of PTCy, cyclosporine A, and mycophenolate mofetil for all cases. High-resolution typing for HLA-A, -B, and -C loci was carried out for all donor and recipient pairs by next-generation sequencing using Omixon Holotype HLA (Omixon, Budapest, Hungary). All patients and donors provided written informed consent for their data to be collected in the PROMISE database of the European Society for Blood and Marrow Transplantation. This study complied with the Declaration of Helsinki and was approved by the Ethics Review Board of Nantes University Hospital.

The clinical outcome and immune reconstitution of some patients have been previously reported [44,45] and were updated in December 2022 for this study. The main objective was to assess the impact of non-expressed vs. expressed donor KIR3DL1 allotype on relapse incidence.

### 2.3. KIR Genotyping

Generic KIR typing was performed for all blood donors (N = 278) and HSC donors (N = 186) using a KIR multiplex PCR-SSP method [46]. KIR genotypes and tel motifs were assigned as reported [8,47]. TelAA, telAB, and telBB KIR motifs in all blood donors were defined, taking into account KIR3DL1/S1/2DS1/2DS4 genes [8]. In particular, telAA individuals were characterized by the presence of KIR3DL1 and 2DS4 and the absence of KIR3DS1 and 2DS1 genes. TelAB individuals were characterized by the presence of KIR3DL1 and 2DS4 with 3DS1 and/or 2DS1 genes. TelBB individuals were characterized by the presence of KIR3DS1 and/or 2DS1 and the absence of KIR3DL1 and/or 2DS4 genes.

### 2.4. KIR Allele Typing

To assign KIR3DL1/S1 alleles in blood donors (N = 278) and HSC donors (N = 186), KIR genes were captured by long-range PCR and subjected to sequencing on a MiSeq sequencer (Illumina, San Diego, CA, USA) after library preparation as reported [38]. KIR3DL1/S1 allele assignment was performed by using Profiler software version 2.24, developed by M. Alizadeh (Research Laboratory, Blood Bank, Rennes, France) [38]. An updated KIR allele library, available on the IPD-KIR database, was implemented in Profiler. KIR3DL1/S1 allele combinations and corresponding tel motifs in blood donors are shown in Appendix A.

### 2.5. KIR3DL1 Constructs

KIR3DL1 constructs were made from pcDEF3-3DL1*004 and pcDEF3-3DL1*002 (control) vectors, kindly provided by P. Parham (Stanford, CA, USA), in which enhanced green fluorescent protein (eGFP) was attached to the C terminus of KIR3DL1 (KIR3DL1-eGFP). Due to unexpected point mutations in the pcDEF3 vector and eGFP, a recombinant PCR approach was used to make chimeric KIR3DL1-eGFP constructs from the pcDEF3-3DL1*004 vector and targeted 3DL1 mutations described in the “Site-directed mutagenesis of KIR3DL1” section. To generate the KIR3DL1-eGFP constructs, the NEBuilder Hifi DNA Assembly cloning kit with Q5 High-Fidelity DNA polymerase (New England Biolabs, Ipswich, MA, USA) was used. Amplification of KIR3DL1 was performed from the pcDEF3-KIR3DL1*004 vector with the sense KIR3DL1 primer (5′-cagatatccatcacactggcccaccatgtcgctcatggtcgtc-3′), which overlaps the 3′ end of pcDEF3, and the antisense KIR3DL1 primer (5′-tgctcaccattgggcaggagacaactttg-3′), which overlaps the 5′ end of eGFP. The amplification of eGFP was performed using the sense eGFP primer (5′-ctcctgcccaatggtgagcaagggcgag-3′), which overlaps the 3′ end of KIR3DL1, and the antisense eGFP primer (5′-acactatagaatagggccctttacttgtacagctcgtccatg-3′). The template for eGFP amplification was the pcDEF3-KIR3DL1*004 vector. Overall, recombinant amplification was performed using the KIR3DL1 and eGFP amplicons as a template with forward KIR3DL1 and reverse eGFP primers and cloned into the pcDEF3 vector. The strategy used to generate KIR3DL1-eGFP constructs is shown in Appendix A. KIR3DL1-eGFP constructs were sequenced on an ABI 3730XL instrument (Eurofins Genomics, Ebersberg, Germany). Error-free KIR3DL1-eGFP clones were subcloned into the pcDEF3 expression vector.

### 2.6. Site-Directed Mutagenesis of KIR3DL1

Point mutations in the KIR3DL1*004-eGFP construct were generated using the GeneArt Site-Directed Mutagenesis System (Life Technologies) and oligonucleotide primers containing the relevant mutations, as recommended by the manufacturer. Position 152 of KIR3DL1*004 was changed from A to G, resulting in a Y30C amino acid change (KIR3DL1*019). Position 320 of KIR3DL1*019 was changed from C to T, resulting in an L86S amino acid substitution (KIR3DL1*019^L86S^). Position 607 of KIR3DL1*019 and KIR3DL1*019^L86S^ was changed from T to C, resulting in an S182P amino acid substitution (KIR3DL1*019^S182P^ and KIR3DL1*019^L86S+S182P^, respectively). The full coding sequences of the resulting KIR3DL1*019, L1*019^L86S^, L1*019^S182P^, and L1*019^L86S+S182P^ plasmids were sequenced to confirm the mutations. KIR3DL1 constructs with their corresponding mutations are shown in Table 1.

### 2.7. Obtention of KIR3DL1-eGFP Transfected Jurkat Cell Line

All DNA constructs used for transfection were prepared using a NucleoSpin Plasmid kit (Macherey Nagel, Hoerdt, France) and sequenced, and only error-free clones were used for transfection. The Jurkat T cell line was transfected using KIR3DL1-eGFP constructs containing the full coding sequence of KIR3DL1*002, L1*004, L1*019, L1*019^L86S^, L1*019^S182P^, and L1*019^L86S+S182P^. Briefly, 10 µg of each KIR3DL1-eGFP construct was transfected by electroporation into 5.10^6^ Jurkat cells with one pulse of 140 V, 1000 µF using a Gene Pulser Xcell System (Biorad, France).

### 2.8. Flow Cytometry Analysis

The KIR3DL1/S1^+^ NK cell surface phenotype was determined from PBMCs by 4-color multiparameter flow cytometry (MFC) using the following mouse anti-human mAbs: anti-CD3-PerCP (SK7), anti-CD56-allophycocyanin (B159) (BD Biosciences), anti-KIR3DL1/S1-PE (Z27 clone; Beckman Coulter, Marseille, France), and anti-KIR3DL1-FITC (DX9) (Beckman Coulter, Immunotech). Expression of KIR3DL1 on transfected cells was evaluated on gated eGFP-positive cells by flow cytometry on a FACSCanto II System (BD Biosciences, Le Pont de Claix, France) using AF647-conjugated anti-KIR3DL1 mAb (Z27) and IgG1 isotype control mAb (MOPC-21, Sony, San Jose, CA, USA). The KIR3DL1-eGFP-transfected Jurkat cell line was permeabilized using cytofix/cytoperm solution (Becton Dickinson, Franklin Lakes, NJ, USA) to determine intracellular KIR3DL1 expression. Cells were stained with APC-conjugated anti-KIR3DL1 mAb (clone 177407; R&D Systems, Minneapolis, MN, USA) and IgG1 isotype control mAb (clone 11711; R&D Systems). KIR3DL1*019^L86S+S182P^-transfected NKLs were pre-incubated with anti-CD107a-BV421 mAb (H4A3; BD Biosciences) at 37 °C. NKL degranulation was assessed after incubation for 5 h alone (negative control) and with different HLA-B-transfected 221 target cells (E/T ratio = 1:1, 2.5 × 10^6^ cells/well) in a 96-well bottom plate with brefeldin A (Sigma, Lezennes, France) at 10 µg/mL for the last 4 h. MFC data were collected on the FACSCanto II (BD Biosciences) and analyzed with Flowjo^TM^ 10.2 software (LLC, Ashland, OR, USA).

### 2.9. Fluorescence Microscopy Imaging

In parallel, KIR3DL1 and HLA class I expression was evaluated on KIR3DL1-eGFP-transfected cells. Membrane labeling was performed using AF647-conjugated anti-KIR3DL1 mAb (clone Z27.3.7; Beckman Coulter) and AF555-conjugated anti-HLA class I mAb (clone F41-1E3; EFS Nantes, France). Intracellular labeling was performed on cells permeabilized with BD Cytofix/Cytoperm solution (Becton Dickinson, Franklin Lakes, NJ, USA) using the AF647-conjugated anti-KIR3DL1 mAb (clone 177407; R&D Systems), the corresponding IgG1 isotype control (clone 11711; R&D Systems), the AF555-conjugated anti-HLA class I mAb (clone F41-1E3), and the corresponding AF555-IgG1 isotype control (clone MOPC-21). Finally, after 2 perm/wash rounds, cells were kept in 50 µL of BD Cytofix/Cytoperm) for 1 h in the dark. After 4 washes with BD Perm/Wash 1×, cells were put on a slide with one drop of Prolong (Invitrogen, Waltham, MA, USA). All slides were kept at 4 °C in the dark for at least 48 h prior to analysis by fluorescent microscopy on an Invitrogen EVOS^TM^ FL Auto Imaging System (Thermo Fisher Scientific).

### 2.10. Statistical Analyses

All statistical analyses were performed using R version 4.2.2 and GraphPad Prism v6.0 software (San Diego, CA, USA). Median follow-up was estimated with the reverse Kaplan–Meier method. Patient characteristics were compared using the chi-squared test for discrete variables and Student t-test for continuous variables. The clinical outcomes studied were overall survival (OS), defined as the probability of survival, and disease-free survival (DFS), defined as survival with no evidence of relapse, from day 0 of hHSCT. OS and DFS were compared using the log-rank test and Kaplain–Meier graphical representation. Relapse was calculated using cumulative incidence, considering non-relapse mortality (NRM) as a competing risk. Univariate and multivariate analyses were performed using the Cox proportional-hazard model. Factors with a *p*-value of <0.1 by univariate analysis or of interest for the study were included in multivariate analysis. A *p*-value of <0.05 was considered statistically significant.

### 2.11. Quality Management System (QMS)

All procedures were conducted under ISO9001:2015 and in compliance with the Guidance Document on Good In Vitro Method Practices.

## 3. Results

### 3.1. Predominant KIR3DL1*004 Allele and Unusual KIR3DL1*019 Allele Are Associated with KIR3DL1 Null Phenotype on NK Cells

We previously reported a high frequency of harboring of KIR3DL1*004 and, to a lesser extent, L1*019 or L1*054 alleles, that are associated with no KIR3DL1 expression on NK cell surface [35]. Here, KIR3DL1/S1 allele polymorphism was assigned in 278 blood donors and correlated to corresponding KIR3DL1 expression on NK cells (Appendix A). Using NGS technology, 16 KIR3DL1/S1 alleles were identified with a frequency higher than 10% for 3DL1*001, L1*002, L1*004, L1*005, and 3DS1*013 (Figure 1a). KIR3DL1*004 was the most frequent allele (22.3%). In contrast, the frequency of 3DL1*019 was low (1.3%) (Figure 1a). From KIR genotypes, and depending on the presence/absence of KIR3DL1, 3DS1, 2DS1, and 2DS4 genes, tel motifs were further defined. The cohort included 167 telAA, 98 telAB, and 13 telBB individuals. Ten individuals (3.6%) harbored two KIR3DL1 alleles and were also 3DS1+ (Appendix A), in accordance with multiple copies of the KIR3DL1/S1 gene already identified in an Irish population [48]. Predominant telAA individuals (60%), who were all 3DL1+/S1−, exhibited high (53%), low (21%), high and low (19%), or no (7%) KIR3DL1 expression on NK cells (Figure 1b). KIR3DL1*004 was present with L1*001, L1*002, L1*015, and L1*008 alleles, leading to high expression on NK cells (Figure 1b). The most frequent allele combination corresponded to 3DL1*004 associated with L1*001, present in 24 individuals (Appendix A). In contrast, the combination of 3DL1*004 with L1*005, L1*007, L1*009, or L1*069 allele led to low KIR3DL1 NK cell expression. Individuals with two copies of 3DL1*004 or L1*019, and had L1*004 associated with L1*019 were characterized as having no KIR3DL1 expression on NK cells. TelAB individuals (35%), who were 3DL1+/S1+, except for the 7 donors who were L1+/S1− but also 2DS1+, exhibited high (50%), low (19%), high and low (1%), or no (30%) KIR3DL1 NK cell expression (Figure 1b). In telAB individuals, 3DL1*004 was associated with L1*001/*002 alleles, leading to high KIR3DL1 expression, or with 3DS1*013, S1*010, or L1*019, leading to a null KIR3DL1 phenotype. In telAB individuals, the most frequent allele combination corresponded to 3DL1*004 associated with 3DS1*013, found in 19 blood donors (Appendix A). Limited to a few individuals, 3DL1*019 was only found associated with L1*004 or S1*013 (Figure 1b). TelBB individuals (5%) were all 3DL1−/S1*013+ (Figure 1b). In a previous study [35], such donors were misidentified as 3DL1*054+/S1+ by a PCR-based sequencing method, which shows the robustness of our NGS approach.

### 3.2. Intracellular Localization of KIR3DL1*019

The KIR3DL1^+^ NK cell phenotype in L1*019-positive individuals suggests a poor expression of the putative protein on the NK cell surface, as reported for L1*004 [27]. Indeed, the expected mature L1*019 protein contains the same L86 and S182 amino acids involved in the intracellular retention of the L1*004 allotype [27], and only one amino acid in the D0 domain differs between L1*004 and L1*019 (Table 1).

The absence of KIR3DL1 on the NK cell surface using DX9 or Z27 binding for L1*019-positive individuals could be due to either a lack of recognition of epitopes targeted by these anti-KIR3DL1 mAbs or the translation of L1*019, impairing its expression on NK cells. To address these hypotheses, we prepared constructs encoding chimeric proteins in which enhanced GFP was attached to the C terminus of KIR3DL1 containing the coding sequence of L1*004 (no Z27 binding) or L1*002 (high Z27 binding). We performed site mutagenesis on L1*004 constructs to evaluate their impact on L1*019 expression (Table 1). The Jurkat cell line was stably transfected with different KIR3DL1-eGFP constructs. Surface and intracellular KIR3DL1 expression were examined using flow cytometry and fluorescent microscopy focusing on eGFP+ cells, since the detection of eGFP was linked to the complete translation of the associated KIR3DL1. As expected, the Jurkat cell line transfected with L1*002 showed high Z27 surface staining and with L1*004 showed no surface staining (Figure 2a,b). The Jurkat cell line transfected with L1*019 also showed no Z27 surface binding (Figure 2a,b). Amino acid substitution at position 182 in the D1 domain (^S182P^) had no effect on Z27 surface binding (Figure 2a,b). In contrast, amino acid substitution at position 86 in the D0 domain alone (^L86S^) restored Z27 surface binding, and more significantly when coupled with amino acid substitution at position 182 in the D1 domain (^L86S+S182P^) (Figure 2a,b). The Jurkat cell line transfected with L1*002 or L1*004 showed intracellular expression of KIR3DL1, although lower expression was observed for L1*004, suggesting either lower binding with the 177,407 mAb or reduced intracellular expression linked to the specificity of this L1*004 allele (Figure 2c). The Jurkat cell line transfected with L1*019 showed intracellular binding comparable to L1*004 (Figure 2c). Amino acid substitution at position 182 in the D1 domain (^S182P^) partially increased the intracellular binding with the 177,407 anti-KIR3DL1 mAb (Figure 2c). Strikingly, amino acid substitution at position 86 in the D0 domain alone (^L86S^) or coupled with amino acid substitution at position 182 in the D1 domain (^L86S+S182P^) strongly increased intracellular 177,407 binding (Figure 2c). Merged images showed co-localization of KIR3DL1 and HLA class I for L1*002, but not for L1*004 (Figure 2d), confirming the intracellular retention of L1*004 [27]. Intracellular staining using the 177,407 mAb revealed retention of L1*019 in the cytoplasm (Figure 2d). Concordant with the flow cytometry data, amino acid substitution at position 182 in the D1 domain (^S182P^) had no effect on KIR3DL1 intracellular expression (Figure 2d).

### 3.3. KIR3DL1*019^L86S+S182P^ NKL Cell Line Recognizes HLA-B-Transfected 221 Target Cells Differently Depending on the Nature of HLA-B Allotypes

Intracellular retention of L1*019 suggests a possible interaction with Bw4^+^ molecules. To test this hypothesis, we evaluated the ability of KIR3DL1*019 ^L86S+S182P^ to modulate the degranulation of NK cells against different HLA-B-transfected 221 cells. Thus, we chose HLA-B-transfected 221 target cells expressing Bw4 (HLA-B*15:13 and -B*38:01) and Bw6 (HLA-B*35 and -B*39:01) molecules. HLA-B*35 and -B*15:13 molecules present the VTAPRTVLL (-21T) leader peptide, and HLA-B*39:01 and -B*38:01 molecules present the VMAPRTVLL (-21M) leader peptide. Of note, only the VMAPRTVLL leader peptide leads to HLA-E membrane expression and CD94/NKG2A binding. This point is important, as NKL expresses the CD94/NKG2A heterodimer which contributes to the inhibition of NKL degranulation. HLA-B was well expressed on all HLA-B-transfected 221 target cells, compared to HLA class I deficient 221 cells (Figure 3a). Although 221-HLA-B*38:01 and 221-HLA-B*39:01 harbor -21M leader peptide, HLA-E expression is highly expressed only on 221-HLA-B*39:01 cells. The degranulation of KIR3DL1*019 ^L86S+S182P^ NKL was determined after 5 h incubation with HLA-B*35, -B*15:13, -B*39:01, and -B*38:01 transfected 221 cells (Figure 3b). Differences in the recognition of HLA-B allotypes were observed. HLA-B*35 and HLA-B*15:13 molecules do not provide good peptide leader to ensure CD94/NKG2A binding with HLA-E ligand, which triggers KIR3DL1*019^L86S+S182P^ NKL degranulation. However, 221-B*15:13 triggers less KIR3DL1*019^L86S+S182P^ NKL degranulation, as it is a Bw4 molecule and is probably recognized by L1*019. In contrast, the -21M leader peptide present in HLA-B*39:01 and HLA-B*38:01 molecules favors the inhibitory signal mediated by CD94/NKG2A and HLA-E interactions. Compared to HLA-B*39:01 (Bw6) molecules, HLA-B*38:01 (Bw4) molecules seem to be better recognized by L1*019, to strongly inhibit KIR3DL1*019 ^L86S+S182P^ NKL degranulation (Figure 3b).

### 3.4. Non-Expressed KIR3DL1 Alleles Are a Risk Factor for Relapse Incidence after T-Replete Haploidentical HSCT in Myeloid Diseases

Our functional data, obtained in vitro using dedicated KIR3DL1 constructs and NKL transfectants, suggest a putative role of KIR3DL1^null^ allotypes. We hypothesized that, in vivo, an inflammatory context, such as the cytokine storm observed early after hHSCT, could favor a specific environment suitable for stabilizing KIR3DL1^null^ allotypes on the NK cell surface. Moreover, we previously reported that KIR genetics impact hHSCT outcomes [44,45], but so far KIR3DL1 allele polymorphism has not been investigated in this context.

Therefore, we investigated the impact of KIR3DL1 alleles in donors on the outcome of hHSCT (N = 186). Using our NGS technology, KIR3DL1 alleles were fully assigned in 166 HSC donors, including 22 donors with non-expressed KIR3DL1 (i.e., homozygous L1*004 or L1*004/3DS1, or L1 negative) and 144 donors with expressed KIR3DL1 (Figure 4a). The clinical characteristics of recipients were comparable in the two groups, based on donor KIR3DL1 expression (Appendix A). The cumulative incidence of relapse after hHSCT in non-expressed vs. expressed KIR3DL1 allotypes did not differ when all patients were included (Figure 4b and Appendix A) or restricted to patients treated for a lymphoid neoplasm (Figure 4b and Appendix A). Interestingly, in patients treated for a myeloid malignancy, the cumulative incidence of relapse was significantly higher in those who received a graft with a non-expressed vs. expressed KIR3DL1 allotype (2 y relapse rate: 47 ± 14% vs. 24 ± 4%, *p* = 0.023) (Figure 4b,c). The same trend was observed in AML patients, although it was not significant due to a limited sample size (Figure 4b and Appendix A). A significant impact of donor KIR3DL1 allotypes on OS or DFS was not found. Based on the hypothesis that KIR3DL1^null^ can be expressed on NK cell surface following stress or stimulus, we evaluated KIR3DL1^null^ expression post-hHSCT (Figure 4d). KIR3DL1 expression on donor NK cells was recovered on recipient NK cells at days 30 and 60. KIR3DL1 is co-expressed with KIR2DL1/2/3. KIR3DL1^null^ is often co-expressed with KIR3DS1, the expression of which is increased after hHSCT, and co-expressed with KIR2DL1/2/3 (Figure 4d). Although NK cells are activated early after hHSCT, KIR3DL1^null^ is not expressed on NK cells, as observed in one representative KIR3DL1^null^ KIR3DS1^+^ HSC donor (Figure 4e).

To determine whether HSC grafts with non-expressed KIR3DL1 increase relapse incidence after hHSCT in patients with myeloid malignancies, univariate and multivariate analyses were performed considering confounding factors, such as age, diseases, DRI, status at the time of transplantation (first complete response (CR) versus subsequent CR versus lack of CR), and conditioning. In myeloid patients, univariate analysis identified recipient age (continuous), diseases (AML versus other myeloid diseases), DRI, status, conditioning, and donor non-expressed KIR3DL1 allotype as significant factors predicting relapse (Table 2). Multivariate analysis confirmed that age, diseases, status, and donor non-expressed KIR3DL1 allotype were associated with relapse after hHSCT in patients with myeloid disease (Table 2). Overall, these results sustain a deleterious effect of donor non-expressed KIR3DL1 alleles on relapse incidence after hHSCT only in the presence of myeloid malignancies.

## 4. Discussion

The KIR3DL1/S1 allele frequencies established here are concordant with previous studies mainly performed in non-European populations [37,49,50]. In particular, we confirm the predominance of the non-expressed L1*004 allele and the scarcity of the KIR3DL1*019 allele in European populations [51,52]. Interestingly, the observed frequency of L1*004 is particularly high, reaching more than 20%. This is in agreement with Alicata et al. [14], who reported that around 20% of individuals had no expressed KIR3DL1 in a huge cohort of unrelated HSC donors from Italy. In addition to L1*004, we report frequencies ranging from 10 to 20% for L1*005, L1*001, L1*002, and S1*013. Of note, these KIR3DL1/S1 alleles are common in Europe [37]. In contrast, in non-European populations, as was recently described for Iranians [53], L1*001 and S1*013 allele frequencies reach near 50 and 30%, respectively, counterbalanced by a low frequency of L1*004 allele, around 5%. Overall, our mechanistic findings, established from French healthy blood donors, highlight a great diversity of KIR3DL1 allele polymorphism depending on telomeric KIR motifs. One could expect that this KIR3DL1 allele polymorphism could impact both NK cell phenotype and functions, as we previously reported for KIR2DL [45].

We showed that L1*019 is retained within the cell but is able to recognize some Bw4 ligands. Substitution of leucine for serine at position 86 in the D0 domain seems to be responsible for the poor folding of L1*019, with a minor contribution from position 182 in D1, as reported for L1*004 [27]. However, both substitutions (L86S and S182P) restored strong L1*019 expression on the membrane of the Jurkat cell line. Intracellular staining revealed poor L1*019 expression in the Jurkat cell line in comparison to the L1*004 allotype. Substitution (L86S) in D0 restored strong intracellular L1*019 staining similar to the L1*002 allotype. Position 30 in the D0 domain constitutes a unique divergent residue between L1*004 and L1*019 that may explain the difference in intracellular staining. We suggest that a functional interaction of KIR3DL1*019^L86S+S182P^ with some HLA-B allotypes expressed on transfected 221 target cells confers an inhibition of NKL degranulation. This result underlines the biological relevance of functional KIR3DL1 allotypes with intracellular localization. KIR3DL1*004 was previously shown to be involved in NK cell licensing, since the small amount reaching the surface can deliver inhibitory signals [28], although inefficient folding causes most of the L1*004 protein to be retained within the cell. The role of L1*019 in NK cell education remains unknown and needs further investigation. Altogether, these observations suggest a potential role for these KIR3DL1 allotypes in a pathological context. In the hHSCT context, NK cells are particularly activated; however, our investigation demonstrates an absence of KIR3DL1^null^ allotype expression on NK cells. We can hypothesize that self-induced or non-self proteins may interact and stabilize KIR3DL1^null^ allotypes on the cell surface, contributing to the NK cell response.

The relevance of polymorphic KIR genes remains a key component of NK cell-based immunotherapies for leukemic patients [54]. The rise of hHSCT in recent years also offers a privileged context to set a beneficial NK cell alloreactivity. KIR3DL1 has been previously correlated with HSCT outcomes [14,15,33], and we demonstrate for the first time that the KIR3DL1 polymorphism may play a pivotal role in relapse incidence after hHSCT. The deleterious effect of donor non-expressed KIR3DL1 allotypes on relapse incidence was only relevant in a limited cohort of patients with myeloid diseases confirming the lineage-specific relapse prediction after hHSCT [55]. Conversely, KIR3DL1*004 is reported to be protective against relapse for patients with AML after HLA-matched HSCT in a large amount of registry data [15]. The divergent effect of this common KIR3DL1 null allele between hHSCT- and HLA-matched HSCT could be related, in particular, to the HLA class I environment [14,56]. In our cohort, HLA-Bw4 environment of donors and recipients defined from both HLA-A and HLA-B typing has been assigned. We did not observe a significant impact of Bw4 environment and/or KIR3DL1/HLA-B subtype combinations on relapse incidence after hHSCT probably due to a limited size cohort. This should be investigated on a larger cohort. The divergent effect of the KIR3DL1*004 allele on HSCT outcome described by Boudreau et al. [15] and that reported here could also be due to the sample size and the heterogeneity concerning the proportion of AML patients. Indeed, the deleterious effect of non-expressed KIR3DL1 alleles on relapse incidence we reported after hHSCT, was observed in a limited cohort of patients with various myeloid diseases. Nonetheless, further investigations on a broader cohort of hHSCT restricted to AML patients would be necessary. In contrast to previous studies [55,57,58], the reported beneficial effect of inhibitory KIR ligand mismatches on relapse incidence after hHSCT did not reach significance here. Moreover, the protective effect of donor cenAA KIR motifs on relapse incidence that we observed in a limited cohort of hHSCT patients [45] was not confirmed, although a trend of less relapse in cenAA than cenB+ donors was observed in patients with myeloid diseases. Differences in the proportions of myeloid patients, conditioning regimens, and stem cell sources between published studies and what is reported here could explain these discordances. In addition to KIR3DL1, the lack of CR was the most significant factor affecting relapse incidence post-hHSCT, as expected [45,59,60].

For patients lacking Bw4, KIR3DL1-expressing NK cells from Bw4+ donors could be alloreactive following hHSCT. Given the predominance of the KIR3DL1*004 allele, the KIR3DL1^+^ NK cell repertoire post-hHSCT could be skewed. The lack of KIR3DL1 expression on NK cells could be associated with an over-representation of the KIR2DL^+^ NK cell compartment, a possibility that needs further investigation. Indeed, donor KIR3DL1^+^ and KIR2DL^+^ NK cell recovery at day 30 post-hHSCT was inversely impacted by KIR ligand mismatches [44]. More broadly, other KIR allele polymorphisms besides KIR3DL1 allotypes that also impact NK cell phenotype and function, such as KIR2DL and KIR2DS4, may be involved after hHSCT and should be investigated in a larger cohort.

## 5. Conclusions

Deciphering KIR allele polymorphism to better characterize the structure of the functional NK cell repertoire remains a significant challenge. The influence of KIR alleles on hHSCT outcomes is still poorly understood. We might expect that knowledge of how KIR allele distributions depend on KIR gene content could help in defining an algorithm to better select haploidentical donors, as reported in HLA-matched unrelated HSCT [8], as well as promoting a beneficial anti-leukemic effect driven by NK cells. More broadly, including KIR allele polymorphism could pave the way to improving our understanding of heterogeneous NK cell responses against acute leukemia [61] and the efficiency of NK-cell-based immunotherapies.

## Figures and Tables

**Figure 1 cancers-15-02754-f001:**
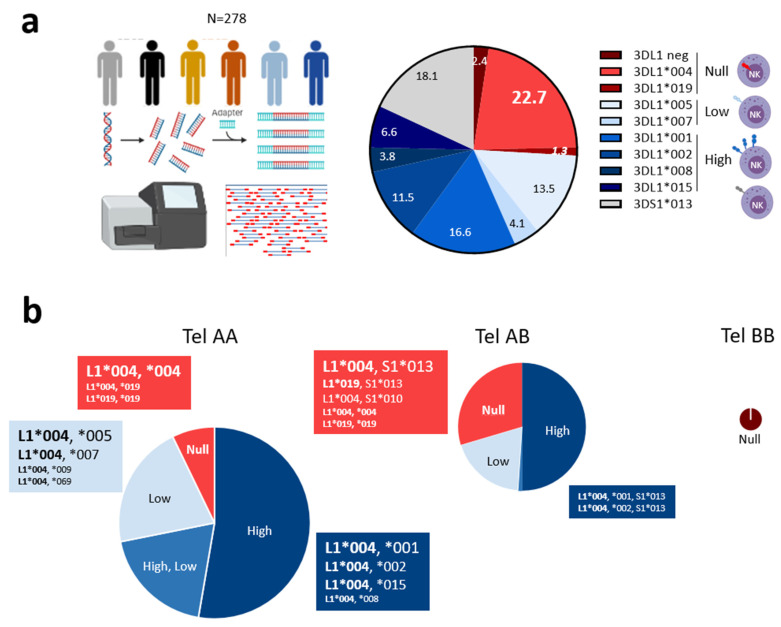
Predominant KIR3DL1*004 allele and unusual KIR3DL1*019 allele are associated with KIR3DL1 null phenotype on NK cells. (**a**) Description of next-generation sequencing (NGS) technology used to type KIR alleles in 278 blood donors, and chart illustrating main KIR3DL1/S1 allele frequencies and KIR3DL1*019 allele (in italics). KIR3DL1/S1 alleles were classified based on corresponding expression on the NK cell surface (null, low, high) using a specific red and blue color gradient code for KIR3DL1 and a gray code for KIR3DS1. (**b**) Pie charts illustrating the frequency of KIR3DL1 expression encountered in telAA (N = 167), telAB (N = 98), and telBB (N = 13) individuals. TelAA individuals were characterized by the presence of KIR3DL1 and 2DS4 and the absence of KIR3DS1 and 2DS1 genes. TelAB individuals were characterized by the presence of KIR3DL1 and 2DS4 with 3DS1 and/or 2DS1 genes. TelBB individuals were characterized by the presence of KIR3DS1 and/or 2DS1 and the absence of KIR3DL1 and 2DS4 genes. Only KIR3DL1*004 and KIR3DL1*019 allele combinations are shown. Pie chart size is proportional to the number of individuals. Font size is proportional to number of individuals with corresponding KIR3DL1/S1 allele combinations in each subgroup.

**Figure 2 cancers-15-02754-f002:**
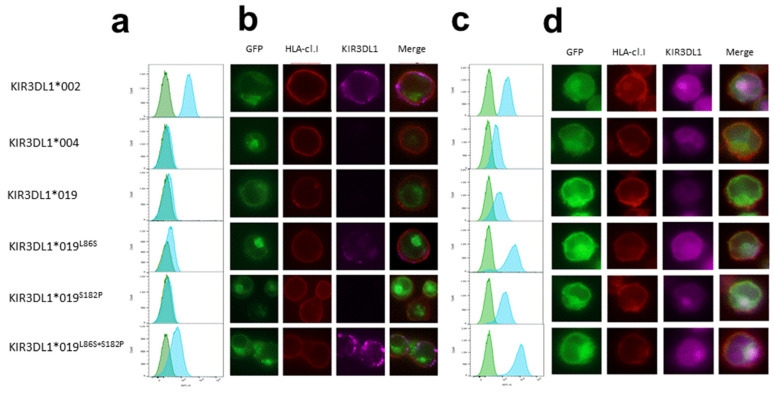
Intracellular localization of KIR3DL1*019. Flow cytometry analysis of Jurkat cell line after stable transfection with KIR3DL1*002, *004, *019, *019^L86S^, *019^S182P^, and *019^L86S+S182P^ chimeric constructs containing eGFP. Only cells positive for eGFP fluorescence were examined. (**a**) Specific Z27 3.7 anti-KIR3DL1 mAb (blue peak; Beckman Coulter) with isotype MOPC-21 IgG control mAb (green peaks; Sony) were used in extracellular staining; (**c**) specific 177,407 anti-KIR3DL1 mAb (blue peak; R&D Systems) with isotype 11,711 IgG control (green peaks; R&D Systems) were used in intracellular staining. Fluorescent microscopy of Jurkat cells stably transfected with KIR3DL1*002, *004, *019, *019^L86S^, *019^S182P^, and *019^L86S+S182P^ chimeric constructs containing eGFP. F41-1E3 anti-HLA class I mAb (EFS Nantes) was used in (**b**) extracellular and (**d**) intracellular transfectant staining. Magnification 100×. Specific Z27 3.7 (Beckman Coulter) and 177,407 (R&D Systems) anti-KIR3DL1 mAbs were used in extracellular and intracellular staining, respectively. Merged images show co-localization of KIR3DL1-eGFP, HLA class I delimiting plasma membrane and KIR3DL1 in extracellular and intracellular staining.

**Figure 3 cancers-15-02754-f003:**
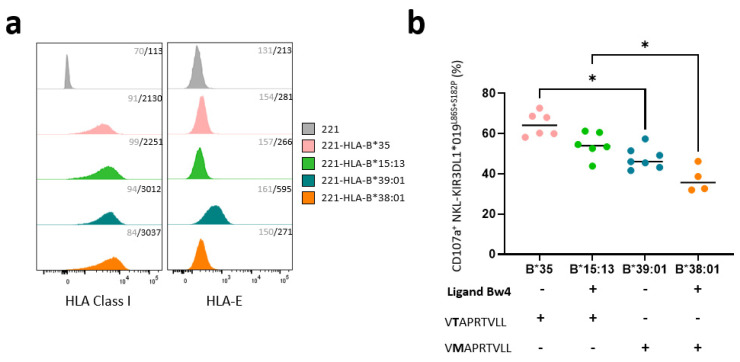
KIR3DL1 *019^L86S+S182P^ NKL cell line recognizes different HLA-B transfected 221 target cells depending on the nature of HLA-B allotypes. (**a**) Histograms showing HLA class I (W6.32 mAb) and HLA-E (3D12 mAb) expression on HLA class I negative 221 cells and four HLA-B transfected 221 cells (HLA-B*35, -B*15:13, -B*39:01, and -B*38:01) determined by flow cytometry analysis. Staining was replicated three times. Control isotype mAb was used to determine negative signal for all 221 targets. Geometric mean fluorescent intensity is indicated in gray for isotype control and black for specific mAbs. (**b**) Degranulation of membrane KIR3DL1*019 ^L86S+S182P^ NKL was determined by measuring CD107a expression after 5 h incubation with HLA-B*35, -B*15:13, -B*39:01, and -B*38:01 transfected 221 cells. Bw4 serological profile of HLA-B molecules and their leader peptide sequence for HLA-E expression are indicated. Only VMAPRTVLL leader peptide led membrane HLA-E expression and CD94/NKG2A binding. Statistical differences between groups were analyzed using one-way ANOVA; * *p* < 0.05.

**Figure 4 cancers-15-02754-f004:**
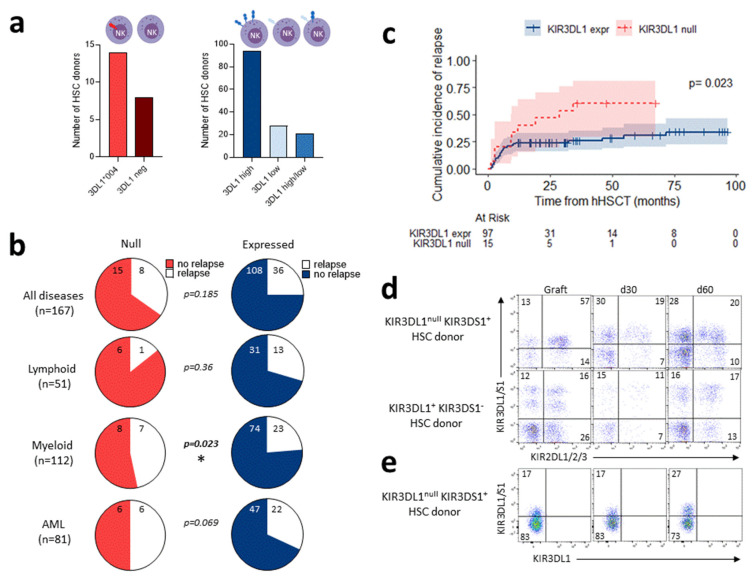
Non-expressed KIR3DL1 alleles are a risk factor for relapse incidence after T-replete haploidentical HSCT in myeloid diseases. (**a**) Representative number of HSC donors with non-expressed KIR3DL1, including homozygous KIR3DL1*004-positive (N = 14) and KIR3DL1-negative (N = 8), compared to HSC donors with expressed KIR3DL1, including high (N = 94), low (N = 28), and high/low (N = 21) allotypes. (**b**) Charts showing relapse incidence at 2 years post-hHSCT performed with HSC donor non-expressed (left panel) and expressed (right panel) KIR3DL1 for all patients (N = 167); and for patients with lymphoid diseases (N = 51), myeloid diseases, excluding 4 idiopathic acquired aplastic anemia (IAA) patients (N = 112), and acute myeloid leukemia (AML) (N = 81). (**c**) Cumulative relapse incidence after T-replete haploidentical HSCT for patients with myeloid diseases grafted with HSC donor non-expressed (N = 16) and expressed (N = 100) KIR3DL1. (**d**) Representative density plots of NK cells expressing KIR3DL1 and KIR3DS1 (Z27 mAb) co-expressed with KIR2DL1/2/3 (GL183 and EB6 mAbs) in graft and at days 30 and 60 post-hHSCT from one KIR3DL1^null^ KIR3DS1^+^ and one KIR3DL1^+^ KIR3DS1^−^ HSC donor. (**e**) Representative density plots of NK cells stained with KIR3DL1/3DS1^−^ and KIR3DL1-specific mAbs (Z27 and DX9 mAbs, respectively), leading to discrimination of KIR3DL1 and KIR3DS1 expression in graft and at days 30 and 60 post-hHSCT from one KIR3DL1^null^ KIR3DS1^+^ HSC donor. Frequency is indicated for each gate. KIR3DL1 alleles are classified depending on corresponding expression on NK cell surface (null, low, high) using a specific red (null) and blue/purple (expressed) color gradient code. * *p* < 0.05.

**Table 1 cancers-15-02754-t001:** KIR3DL1 constructs *.

DOMAIN	Lea.		D0							D1		D2	Stem	Trm	Cytop.
POSITION	−20	−9	2	30	31	44	47	54	86	182	238	283	320	343	373
KIR3DL1*002	L	F	V	Y	R	R	V	L	S	P	R	W	I	C	E
KIR3DL1*004	S	L	M	Y	H	G	I	I	L	S	G	L	V	Y	Q
KIR3DL1*019	S	L	M	**C**	H	G	I	I	L	S	G	L	V	Y	Q
KIR3DL1*019^L86S^	S	L	M	**C**	H	G	I	I	**S**	S	G	L	V	Y	Q
KIR3DL1*019^S182P^	S	L	M	**C**	H	G	I	I	L	**P**	G	L	V	Y	Q
KIR3DL1*019^L86S+S182P^	S	L	M	**C**	H	G	I	I	**S**	**P**	G	L	V	Y	Q

* Amino acid substitutions that distinguish KIR3DL1 allotypes. All KIR3DL1*019 constructs are compared with reference KIR3DL1*002 and 3DL1*004 constructs. Amino acid changes between KIR3DL1*004 and 3DL1*019 and specific point mutations are shown in bold and corresponding positions are shown shaded in gray. Lea., leader; Trm, transmembrane; Cytop, cytoplasmic.

**Table 2 cancers-15-02754-t002:** Univariate and multivariate analysis of variables affecting relapse after T-replete haploidentical HSCT using post-transplant cyclophosphamide in patients with myeloid diseases *.

			Univariate Analysis		Multivariate Analysis	
Variable		n (%) **	HR (95% CI)	*p*-Value	HR (95% CI)	*p*-Value
Age		105 (100%)	**0.96 (0.94–0.98)**	**0.001**	**0.97 (0.94–1.00)**	**0.02**
Diseases	AML	77 (73.3%)	Reference		Reference	
	Other myeloid	28 (26.7%)	**0.34 (0.14–0.82)**	**0.02**	**0.24 (0.06–0.95)**	**0.04**
DRI	Low–intermediate	57 (54.3%)	Reference		Reference	
	High–very high	48 (45.7%)	**4.79 (2.31–9.96)**	**<0.001**	2.60 (0.91–7.44)	0.08
Status	CR = 1	48 (45.7%)	Reference		Reference	
	CR > 1	10 (9.5%)	**3.09 (1.12–8.53)**	**0.03**	1.35 (0.35–5.16)	0.67
	No CR	47 (44.8%)	**3.42 (1.51–7.73)**	**0.003**	**4.58 (1.04–20.20)**	**0.04**
Conditioning	RIC **	83 (79%)	Reference		Reference	
	Sequential	22 (21%)	**3.90 (2.02–7.52)**	**<0.001**	0.42 (0.11–1.70)	0.23
KIR3DL1 allotype	Expressed	91 (86.7%)	Reference		Reference	
	Null	14 (13.3%)	**2.49 (1.24–4.96)**	**0.01**	**2.10 (1.11–3.98)**	**0.02**

HR, hazard ratio; CI, confidence interval; AML, acute myeloblastic leukemia; * other myeloid excluding 4 IAA: myelodysplastic syndrome, myeloproliferative neoplasm, or mixed syndrome; DRI, disease risk index; CR, complete response; RIC, reduced-intensity conditioning ** Two single MAC patients were removed from the analysis.

## Data Availability

The data presented in this study are available on request from the corresponding author.

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
