# Peer review of "Non-Expressed Donor KIR3DL1 Alleles May Represent a Risk Factor for Relapse after T-Replete Haploidentical Hematopoietic Stem Cell Transplantation"

_cancers, 2023, doi:10.3390/cancers15102754_

Round 1

Reviewer 1 Report

Dear authors :

Congratulations on your hard and beautiful work to produce the paper “Non-expressed donor KIR3DL1 alleles are a risk factor for relapse after T-replete haplo-identical hematopoietic stem cell transplantation.”

It is a novel work that must be published but the clinical data is much weaker than the laboratory work due to the relatively small number of patients: it is not be appropriate to make a strong statement and the title of the paper based on only 14 hematopoietic stem cell haploidentical donors with KIR3DL1 null allotype for patients with myeloid malignancies (Table 2), even with a p of 0.02.

The role of KIR3DL1 in acute myelogenous leukemia has been demonstrated in a large registry data with 1,328 patients as published - your reference [15]  Boudreau JE, et al. [KIR3DL1/HLA-B Subtypes Govern Acute Myelogenous Leukemia Relapse After Hematopoietic Cell Transplantation. J Clin Oncol. 2017 Jul 10;35(20):2268-2278. doi: 10.1200/JCO.2016.70.7059. Epub 2017 May 18. PMID: 28520526; PMCID: PMC5501362. ] that concludes that “Consideration of KIR3DL1-mediated inhibition in donor selection for HLA-matched HCT may achieve superior graft versus leukemia effects, lower risk for relapse, and an increase in survival among patients with AML.” HLA-B of the recipients was also studied in this paper, and the combinations were evaluated. You do not mention patients' KIR or HLA-B recipient typing, and it must be mentioned in the discussion.

Aplastic anemia patients might be excluded since it is not a malignant disease, but it would be ok to maintain them.

I would not change the description of the laboratory work to study French healthy blood and hematopoietic stem cell donors, but I missed the discussion and conclusion about these results.

In conclusion, congratulations for your work. I would recommend highlighting your laboratory findings and to change the title of the paper and the conclusion since they are based on only 14 hematopoietic stem cell haploidentical donors with KIR3DL1 null allotype for patients with myeloid malignancies.

*Please use haploidentical (no hyphen) throughout the paper.

English language is fine - haploidentical does not have a hifen (NOT haplo-identical), please correct.

Author Response

Congratulations on your hard and beautiful work to produce the paper “Non-expressed donor KIR3DL1 alleles are a risk factor for relapse after T-replete haplo-identical hematopoietic stem cell transplantation.”It is a novel work that must be published but the clinical data is much weaker than the laboratory work due to the relatively small number of patients: it is not be appropriate to make a strong statement and the title of the paper based on only 14 hematopoietic stem cell haploidentical donors with KIR3DL1 null allotype for patients with myeloid malignancies (Table 2), even with a p of 0.02.

Author response : We agree with the reviewer’s comment. We propose the new title « Non-expressed donor KIR3DL1 alleles may represent a risk factor for relapse after T-replete haploidentical hametopoietic stem cell transplantation ».

The role of KIR3DL1 in acute myelogenous leukemia has been demonstrated in a large registry data with 1,328 patients as published - your reference [15] Boudreau JE, et al. [KIR3DL1/HLA-B Subtypes Govern Acute Myelogenous Leukemia Relapse After Hematopoietic Cell Transplantation. J Clin Oncol. 2017 Jul 10;35(20):2268-2278. doi: 10.1200/JCO.2016.70.7059. Epub 2017 May 18. PMID: 28520526; PMCID: PMC5501362. ] that concludes that “Consideration of KIR3DL1-mediated inhibition in donor selection for HLA-matched HCT may achieve superior graft versus leukemia effects, lower risk for relapse, and an increase in survival among patients with AML.” HLA-B of the recipients was also studied in this paper, and the combinations were evaluated. You do not mention patients' KIR or HLA-B recipient typing, and it must be mentioned in the discussion.

Author response : We agree with the reviewer’s comment. High resolution typing for HLA class I loci was available for all donor/recipient pairs. This point has been added in the Material and Methods section (lines 128-130). The HLA-Bw4 environment of donors and recipients was assigned depending on both HLA-A and HLA-B typing. We did not observe any significant impact of Bw4 environment and/or KIR3DL1/HLA-B combinations on relapse incidence after hHSCT due to a limited sample size. However, the potential role of Bw4 environment and KIR3DL1/HLA-B subtypes on hHSCT outcome should be investigated on a larger cohort. Thanks to this fruitful comment, we improved our discussion on the potential role of HLA-Bw4 environment and KIR3DL1/HLA-B subtypes after hHSCT (lines 515-520). The limitations of our study compared to Boudreau et al. have been also clearly identified (lines 510, 512, 520-526). We previously reported that reconstituting NK cells after T replete haploidentical HSCT (hHSCT) show a donor KIR profile (Willem et al., JI 2019, 202:2141-2152). This point was added in the Discussion section (line 540) Therefore, we think that it is not appropriate to determine patient KIR typing.

Aplastic anemia patients might be excluded since it is not a malignant disease, but it would be ok to maintain them.

Author response : We agree with the reviewer’s comment and we apologize for this misunderstanding. Indeed, idiopathic acquired aplastic anemia (IAA) patients (N = 4) were already excluded in multivariate analysis. To improve the understanding of the impact of non-expressed KIR3DL1 alleles on relapse incidence after hHSCT, we have added this point in the revised manuscript (lines 436-437, 462).

I would not change the description of the laboratory work to study French healthy blood and hematopoietic stem cell donors, but I missed the discussion and conclusion about these results. In conclusion, congratulations for your work. I would recommend highlighting your laboratory findings and to change the title of the paper and the conclusion since they are based on only 14 hematopoietic stem cell haploidentical donors with KIR3DL1 null allotype for patients with myeloid malignancies.

Author response: The title and the conclusion of the manuscript (lines 477-481, 508, 510, 512, 515-526) has been modified accordingly.

*Please use haploidentical (no hyphen) throughout the paper.

Author response : This has been corrected throughout the paper.

Reviewer 2 Report

This manuscript describe very interesting data on KIR polymorphisms in european population.

While the initial data are only descriptive, the authors thereafter well explored KIR polymorphisms with 1/ expression data after transfection experiments 2/ functional degranulation experiments. 

And, then, the explorations were extended to clinical data, and KIR impact on allo-HSCT outcomes. 

Manuscript is interesting, well-written, and presents very convincing data on the importance of KIR polymorphims explorations in the donor before allo-HSCT and before NK cells immunotherapies. 

Author Response

This manuscript describe very interesting data on KIR polymorphisms in european population. While the initial data are only descriptive, the authors thereafter well explored KIR polymorphisms with 1/ expression data after transfection experiments 2/ functional degranulation experiments. And, then, the explorations were extended to clinical data, and KIR impact on allo-HSCT outcomes. 

Manuscript is interesting, well-written, and presents very convincing data on the importance of KIR polymorphims explorations in the donor before allo-HSCT and before NK cells immunotherapies. 

Author response : We thank you the reviewer for his valuable comments.